# Earliest millet cultivation reflects steppe connections, dietary flexibility, and resilience in Bronze Age northern Greece

Kyriaki Karanikola[1]*, Soultana-Maria Valamoti[2], Giedrė Motuzaite Matuzeviciute[1]

1 Department of Archaeology, Faculty of History, Vilnius University, Vilnius, Lithuania, 2 LIRA Laboratory, Department of Archaeology, School of History and Archaeology, Aristotle University of Thessaloniki, Thessaloniki, Greece

* kyriaki.karanikola@if.stud.vu.lt

## Abstract

This paper explores early broomcorn millet (hereafter millet) cultivation in Greece during the Bronze Age. The primary archaeobotanical data for this study derive from the site of Skala Sotiros on the island of Thasos in northern Greece. The site provides unique insights into localized Bronze Age agricultural practices, revealing both divergence from southern Greece agricultural systems and potential influences from exchange networks that linked northern Greece to the southern Balkans and the Pontic steppe–Black Sea region. Systematic sampling of the Bronze Age layers at Skala Sotiros has yielded a diverse assemblage with a notable abundance of millet (*Panicum miliaceum*), a crop almost absent from contemporary southern Greece. Recent radiocarbon dates on millet grains from Skala Sotiros contribute new evidence toward understanding the routes through which millet could have been introduced into the region during the Bronze Age. This study explores the interplay of environmental and cultural factors in the dispersal of millet in Greece, considering environmental stress, cultural dynamics, population movements, and interaction networks. The extensive review of archaeobotanical data across Greece demonstrates how the cultivation of millet may have served as a culinary identity signifier, providing further evidence of differences between northern and southern Greece.

## Introduction

The Bronze Age in Greece was a pivotal period marked by technological innovation, the intensive use of metal alloys, population movements, and the expansion of trade [1,2]. On the Greek mainland, it spanned roughly between 3000 and 1200 BCE. The Bronze Age period is conventionally divided into Early, Middle, and Late phases, although regional variability is reflected in divergent chronological frameworks. Archaeological research over the last decades has shown differences between the north and the south of Greece, the socioeconomic elements of which were very

**Data availability statement:** All relevant data are within the manuscript and its Supporting information files.

**Funding:** This research was funded by the European Union with a Consolidator Grant awarded to Giedrė Motuzaitė Matuzevičiūtė (ERC-CoG, MILWAYS, 101087964). Additional AMS radiocarbon dating of millet grains performed at Poznań was funded by the ERC Consolidator Grant PlantCult (ERC CoG, GA682529, PI Soultana Maria Valamoti). Views and opinions expressed are those of the authors only and do not necessarily reflect those of the European Union or the European Research Council Executive Agency. Neither the European Union nor the granting authority can be held responsible for them. The funders provided financial support for specific laboratory analyses (AMS radiocarbon dating), but had no role in the study design, data analysis and interpretation, decision to publish, or preparation of the manuscript.

**Competing interests:** The authors have declared that no competing interests exist.

thoughtfully discussed more than 30 years ago by Paul Halstead [3]. In northern Greece (Table 1), the Early Bronze Age may have begun earlier than in the south, around 3300/3100 BCE, and appears to represent a long and relatively uninterrupted phase until 2300/2200 BCE, whereas in southern Greece it is subdivided into clearer cultural transitions, starting around 3000 BCE until 2100 BCE [4–7]. Evidence for the Middle Bronze Age in northern Greece, 2300/2200–1700/1550 BCE, remains limited, yet sites such as Assiros, Archontiko, and Ayios Mamas suggest continuity from earlier phases and the appearance of more complex architecture [8,9]. The Late Bronze Age is broadly dated between 1700 and 1200 BCE in the north, while in the south it is divided into multiple, more finely defined phases [6,7]. The subdivision into sub-periods for southern Greece is largely based on the archaeological record from the Mycenaean centers, which characterize the southern Greek mainland during the Late Bronze Age. Based on archaeological evidence, the Mycenaean influence extended from its southernmost limit on the island of Crete to its northern frontier at Mount Olympus, which acted as a boundary with regions that developed along different trajectories [4,10–12].

In this paper, the term "northern Greece" refers to Greek Macedonia, Thrace, and the islands of the North Aegean, all located north of Mount Olympus (Fig 1). In contrast, "southern Greece" includes regions such as the Peloponnese, Central Greece, and Thessaly, where the influence of the Mycenaean palatial centers was more direct and intense. Crete and the Minoan centers are not included in the present study, although they played a significant role in the Aegean during the Late Bronze Age. The focus here is on the relationship between northern Greece and the Mycenaean centers of southern Greece.

The Late Bronze Age marks a significant departure from the preceding centuries, with notable differences observed in mainland Greece's socioeconomic organization and material culture. From the mid-15th to the 12th century BCE, those parts of mainland Greece characterized by centralized economic and political systems, and the featured palatial centers, specialized workshops, and the production of distinct pottery types and other artefacts, as well as monumental funerary architecture that reflects emerging social differentiation [2,13–16]. These political and material developments were closely tied to economic strategies centered on agricultural production and interregional exchange. The economy in major centers—such as Mycenae, Pylos, and Tiryns—focused on agricultural production, storage, and the redistribution of goods. Surplus was exported through trade, forming a well-connected network across mainland Greece and the wider Mediterranean, as evidenced by finds originating from Crete, Cyprus, Anatolia, the Levant, and the western Mediterranean [14,17–21].

On the other hand, northern Greece shows a different picture during the same period. The strong Mycenaean influence appears to extend as far north as Mount Olympus, as demonstrated by the warrior burials and Mycenaean-style weapons (swords, daggers, spearheads) from the cemetery at Spathes [10,11].

Despite indications of contact with Mycenaean centers, Mycenaean influence in northern Greece is only minimally attested and mainly confined to a few ceramic finds

**Table 1.  Archaeological Phases and Chronology for Northern Greece: after Andreou et al. 1996 [4], with the Late Bronze Age chronology updated following recent radiocarbon dating [22–24].**

| NORTH GREECE | CHRONOLOGY |
| --- | --- |
| Early Bronze Age | 3300/3100–2300/2200 BCE |
| Middle Bronze Age | 2300/2200–1700/1550 BCE |
| Late Bronze Age | 1700/1550–1200 BCE |

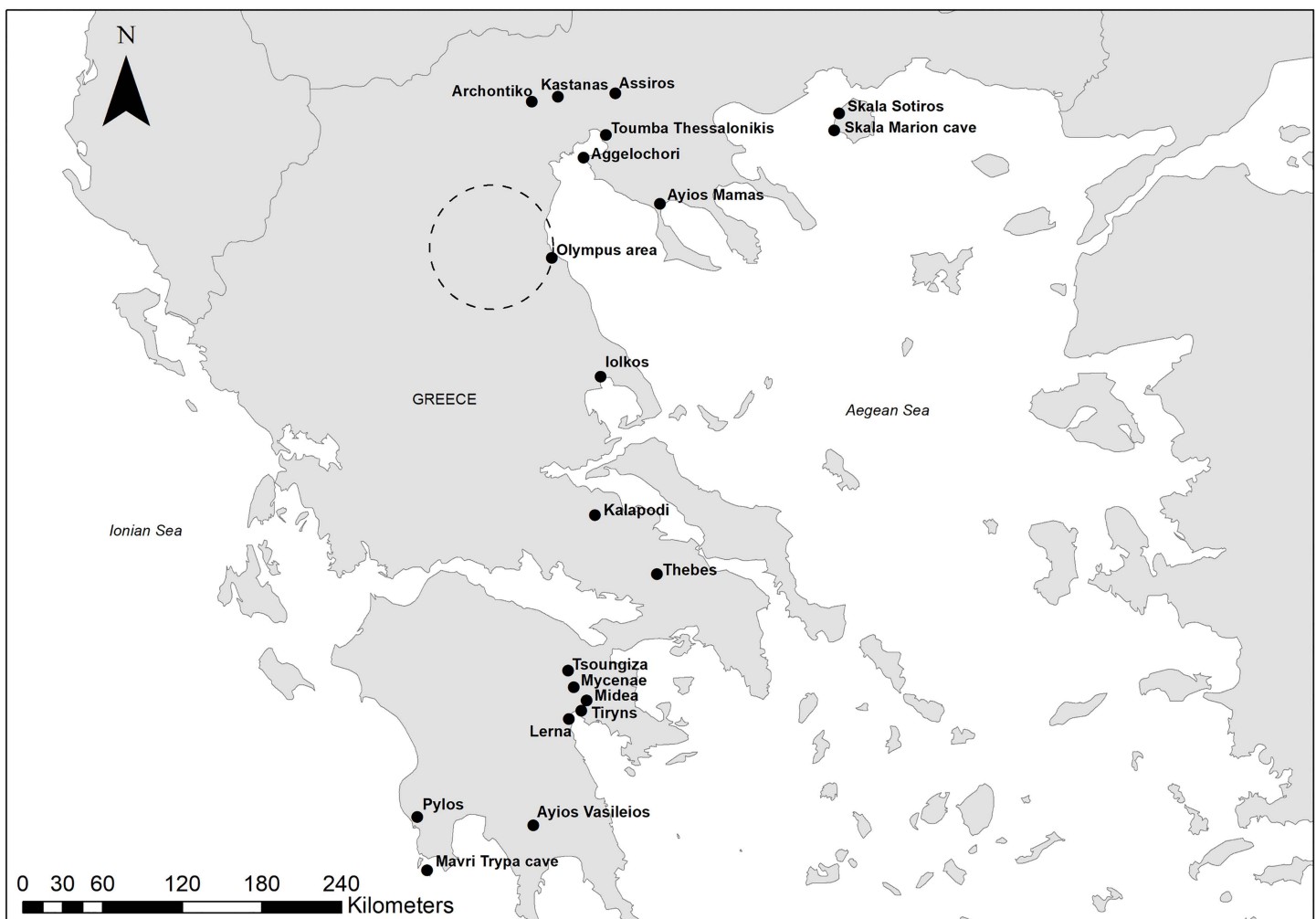

**Fig 1.  Map showing the sites mentioned in the text.** Map compiled in ArcMap 10.8 using a basemap made from Natural Earth.

[12,25,26]. However, recent large-scale research in the eastern Balkans has suggested that at least part of the prestigious objects reaching this region from the Mycenaean centers circulated through northern Greece. This evidence indicates that contacts between northern and southern Greece cannot be assessed solely based on ceramic distributions. Instead, the northern regions seem to have functioned primarily as corridors and mediating zones within the exchange networks that characterized the Bronze Age [27].

The predominant settlement type in northern Greece was the tell, with smaller communities organized around autonomous households and generally less surplus agricultural production compared to the Mycenaean centers [4,25,26]. Some sites, however, such as Assiros, provide archaeobotanical evidence suggesting more extensive agricultural activity and storage [4,26,28,29]. Differences between the north and the south are also observed in some of the cultivated species, although the available data are not of comparable resolution for both the north and the south of Greece [30,31]. Among the cereal crops, emmer wheat (*Triticum dicoccum*) was more prominent in southern Greece already during the Neolithic, whereas einkorn wheat (*Triticum monococcum*) is attested in large stored quantities at the Bronze Age site of Assiros (northern Greece), interpreted as a commodity possibly used in exchange with the south [29]. Additionally, millet (*Panicum miliaceum*) appears with high frequency in archaeobotanical assemblages from Late Bronze Age contexts in northern Greece, while such remains are very rare in the south [3,31,32]. Recent regional syntheses based on presence–absence data confirm the prominence of millet in the north, a pattern further supported by findings from the eastern Aegean and western Anatolia [33].

Oil crops also illustrate this north–south differentiation. Olive (*Olea europaea*), a key oil crop in the Bronze Age, is notably characteristic of southern Greece [34], whereas in the north, plant remains point to a range of other oil plants such as linseed and *Lallemantia* species, suggesting differences in crop choices between the north and southern Greece [35].

## Climate and environmental change during the bronze age

In the period under study (ca. 3000–1200 BCE), two major climatic events have been proposed on the basis of paleoenvironmental and paleoclimatic reconstructions: the 4.2 ka event (ca. 2200 BCE) and the 3.2 ka event (ca. 1200 BCE). Both events have been discussed as episodes of climatic stress that may have affected societies in the Aegean and wider eastern Mediterranean, although recent syntheses emphasize strong regional variability and caution against mono-causal interpretations [36,37].

**The 4.2 ka event (ca. 2200 BCE).** At the onset of the Bronze Age, Psychoyos [38] documented fluctuations in sea level and coastal morphology across the Aegean, highlighting environmental instability that influenced settlement and land use. This interpretation has been corroborated by later paleoclimatic studies linking these shifts to the 4.2 ka event. Jacobson et al. [37] and Hazell [39] identify a marked phase of drought between 2150 and 2000 BCE, while Fouache et al. [40] and Zanchetta et al. [41] report evidence from lake sediment cores in Albania (western Balkans) showing drier conditions around 2800–2000 BCE. Syrides et al. [42] also note that in northern Greece, lowland areas previously dominated by wetlands began to transform into more open and drier landscapes during the Early Bronze Age, a change attributed to a combination of increased aridity and relative sea-level fluctuations.

**The 3.2 ka event (ca. 1200 BCE).** A second phase of aridification is associated with the 3.2 ka event, around 1200 BCE, often linked to the collapse of Bronze Age societies in the eastern Mediterranean. Carpenter [43] was among the first to suggest that climatic change contributed to the decline of the Mycenaean palatial centers, while Bryson et al. [44] and Weiss [45] also emphasized the role of climatic deterioration and drought. Multi-proxy data from Kaniewski et al. [46–48] indicate severe and prolonged droughts in the eastern Mediterranean during the late 13th and early 12th centuries BCE.

In mainland Greece, drought conditions that began shortly before 1200 BCE and peaked around that time are seen as one of several contributing factors—alongside population movements, trade instability, and seismic activity—to the collapse of Mycenaean centers and the onset of the post-palatial period. Supporting evidence includes pollen sequences showing aridification in mainland Greece [46–48], archaeological analyses linking environmental stress with settlement disruption in the Argolida [49], and speleothem records from the Peloponnese documenting severe drought episodes [39]. At the same time, recent paleoclimatic syntheses emphasize that climatic stress during this period was expressed through regionally variable and multi-faceted patterns, combining aridity, cooling, and increased seasonal unpredictability rather than a single uniform signal [36]. The extent to which it acted as a primary driver of social change and contributed to the collapse of the Mycenaean centers, however, remains contested [39,50].

### Regional variation: North versus south

The literature on southern Greece reports multiple indications of climatic changes at the end of the Bronze Age (12th century BCE). By contrast, the situation in northern Greece and the North Aegean is less clear. For instance, oxygen and carbon isotope records from speleothems in Skala Marion Cave (Thasos) show no distinct aridification trend, in contrast to southern Aegean records [39].

Similarly, multi-proxy evidence from the western Thermaic Gulf indicates no abrupt climatic deterioration around 1200 BCE [51]. The pollen record there suggests landscapes shaped primarily by human activity rather than environmental stress. By contrast, palynological data from Tristinika marsh in Halkidiki (north Greece) indicate increased climatic stress toward the end of the 12th century BCE, with a decline in *Typha latifolia* and *Fagus* pollen—both sensitive to water availability—pointing to cooler and drier conditions [52].

Given the complexity of environmental and cultural factors involved in choices relating to agricultural practice and crop choices in particular, this paper, based on the full study of the archaeobotanical assemblage from Skala Sotiros, constitutes an original contribution to the discussion of the factors involved in the introduction of millet in Greece and South-Eastern Europe. The integration of a new archaeobotanical assemblage that shows early millet finds from the region, together with new AMS dates for these finds, provides a solid basis for a detailed examination of the arrival and status of millet as a crop for the Bronze Age societies of Greece. Cultural and environmental proxies for the study region and time period, together with the archaeobotanical finds, offer a unique opportunity to disentangle the cultural and possible environmental factors involved in the introduction of a new cereal crop in South-Eastern Europe. The paper thus aspires to contribute to wider debates on crop dispersal and agricultural adaptation in the Bronze Age Aegean.

## Materials and methods

### The archaeological site

The site of Skala Sotiros is located on the island of Thasos in the North Aegean (Fig 2). Thasos, the northernmost Greek island in the Aegean Sea, lies close to the mainland. Its coastline extends for 115 km, while the land area covers 378.84 km². The island's climate, combined with the presence of gold, iron, and white marble, attracted human settlement already in prehistoric times [53].

Occupation on Thasos is attested since the Neolithic, as indicated by settlements such as Limenaria, Agios Antonios, and Kastrí, as well as Bronze Age sites including Agios Ioannis, Limenaria, Agios Antonios Potos, and Skala Sotiros [54].

The archaeological site lies in the coastal village of Skala Sotiros, named after the site itself, on the hill of Prophet Elias, beneath the village church. In 1985, excavations uncovered a settlement of the Early Bronze Age featuring a distinctive stone enclosure [55]. The first investigations were carried out in 1985 by the Archaeological Service of Kavala. In the uppermost layer, a cemetery of Early Christian to Roman date was identified, alongside a fortified Bronze Age settlement covering an area of ca. 1,350–1,400 m² [55].

Subsequent excavations (1986–1990) revealed two construction phases within the Bronze Age settlement. The earlier phase comprised the enclosure and the buildings within it [54]. Some sections of the wall display herringbone masonry, a technique also known from other Early Bronze Age sites such as Troy I and the eastern Aegean [55].

A semicircular building yielded a floor under which three trenches were opened, exposing storage and tableware pottery as well as small finds, including a distinctive gold ornament with parallels at Troy [56,57]. In addition, seven fragments of anthropomorphic menhir-type stelae were recovered, reused as construction material in the stone enclosure; such stelae are well attested in the Black Sea region and are discussed below [55,58]. These earlier layers were sealed by large fragments of burnt clay, interpreted as collapsed roof remains from a destructive fire [55,56,58].

The second settlement phase appears to have been established directly above this destruction horizon. Finds from this phase were documented in several trenches and indicate continuity from the earlier occupation, with no chronological

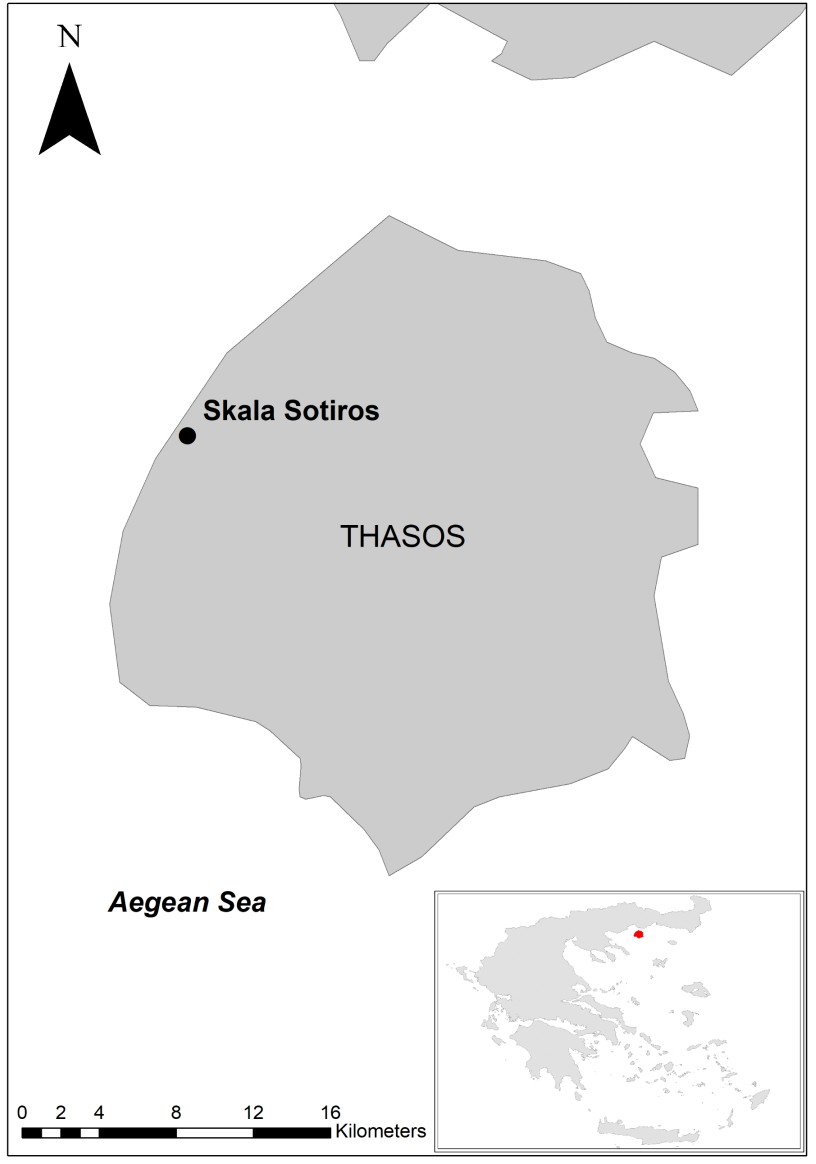

**Fig 2. Map of the island of Thasos showing the archaeological site of Skala Sotiros.** Map compiled in ArcMap 10.8 using a basemap made from Natural Earth.

gap [54,55]. A few sherds of Late Bronze Age pottery suggest that occupation at Skala Sotiros may have continued on a smaller scale into later centuries [54,59].

On the basis of archaeological evidence, three Bronze Age phases have been identified at Skala Sotiros, all attributed to the Early Bronze Age (Table 2). The earliest (EBA I) is represented only by a few finds, including incised pottery sherds and anthropomorphic stelae. Phase II (subphases a–c), beginning around 2600 BCE, marks the initial construction and subsequent modifications of the stone enclosure and is dated to ca. 2600–2000 BCE. Phase III, starting around 2000/1800 BCE, overlies the destruction layer of Phase IIc and shows continued occupation. Although its end remains uncertain, the presence of a few sherds and radiocarbon-dated grains suggests limited continuation beyond this period. A later stratigraphic layer belongs to the historical era and is unrelated to the Bronze Age occupation.

**Table 2. Chronology of the site Skala Sotiros after Koukouli-Chrysanthaki and Papadopoulos 2016 [54].**

| SKALA SOTIROS | CHRONOLOGY |
|---|---|
| Early Bronze Age IIa | 2556−2391 BCE |
| Early Bronze Age IIb | 2391−2223 BCE |
| Early Bronze Age IIc | 2223− 1803 BCE |
| Early Bronze Age III | 1803− 1700 BCE |

## Study methods

All necessary permits were obtained for the described study, which complied with all relevant national and institutional regulations. Archaeological excavation and sampling at Skala Sotiros (Thasos, northern Greece) were originally conducted by the Hellenic Ephorate of Antiquities of Kavala during the 1980s. Permissions for the sampling and export of millet and barley grains from the archaeobotanical assemblage was provided by the Hellenic Ministry of Culture and Sports, Directorate of Conservation of Ancient and Modern Monuments (Permit No. ΥΠΠΟΑ/Φ77/130981, 13 July 2021 and Permit No. ΥΠΠΟ/Φ77/442871π.ε., 11 August 2025). Additional information regarding the ethical, cultural, and scientific considerations specific to inclusivity in global research is included in the Supporting Information (S2 File). The samples were transferred to the Vilnius Radiocarbon Laboratory for AMS radiocarbon dating. All archaeobotanical specimens analyzed in this study are stored at the Aristotle University of Thessaloniki, Greece.

Archaeobotanical samples were collected during the excavations at Skala Sotiros (1986–1987) from a range of archaeological contexts. Sampling targeted both contexts with visible charred plant remains and specific features such as buildings, pottery concentrations, hearths, and burnt layers. Recovery was carried out by water flotation using a variant of the Ankara machine [55]. During the 1986–1987 excavation seasons, cloth rather than fine-mesh sieves was used for the collection of the light fraction, a practice that may have affected the representation of small or fragile plant remains. A total of 80 sediment samples were processed, of which 11 were excluded as belonging to historical rather than Bronze Age contexts. The remaining 69 samples derive from three trenches (TI, TIIA, TIIB) within the enclosure, where a burnt destruction layer was also identified. The archaeobotanical material was initially examined by S.-M. Valamoti and subsequently re-analyzed in the framework of the present study. Since the sediment volumes per sample were not recorded during the 1986–1987 excavations, densities (items per liter) could not be calculated. Instead, results are presented in terms of absolute counts, relative percentages of the identified plant remains, and frequencies of occurrence across the analyzed samples. Based on stratigraphic and contextual information, all 69 samples were assigned to the Early Bronze Age (site chronology): 5 samples to Phase IIa, 11 to Phase IIb, 6 to Phase IIc, and 47 to Phase III. Plant remains were quantified by counting identifiable macroremains such as cereal grains, pulses, seeds, and fruit stones. Each item was recorded as a separate unit only if it was sufficiently preserved to allow taxonomic identification; highly fragmented or unidentifiable material was noted but excluded from quantification. Plant identification was based on comparison with modern reference collections and published atlases. Plant nomenclature follows Plants of the World Online (POWO, Royal Botanic Gardens, Kew), with reference also to *Flora Europaea* for regional distributions [60,61]. A proportion of the remains could not be securely assigned to specific taxa. These were recorded in intermediate or indeterminate categories where preservation or morphology did not allow confident identification. This applies particularly to groupings such as *Lathyrus sativus/ Vicia ervilia*, large legume indet., *Triticum/Hordeum*, *Triticum monococcum/dicoccum*, and cereal indet., reflecting diagnostic limitations due to fragmentation or carbonization.

Percentages presented in the results were calculated in relation to the total number of identified plants remains per phase and assemblage, to illustrate relative abundance alongside absolute counts. The samples were analyzed at the Laboratory for Interdisciplinary Archaeological Research (LIRA) and at the PlantCult Laboratory, Centre for

Interdisciplinary Research and Innovation (CIRI), Aristotle University of Thessaloniki. The remains were examined under a stereomicroscope at magnifications ranging from ×8 to ×80.

The archaeobotanical assemblage from Skala Sotiros comprises a diverse range of cultivated and wild plant taxa (Figs 3 and 4), primarily cereals and pulses, along with a smaller number of wild species. For the purposes of presentation, the material is grouped by excavation sector.

### Radiocarbon dating and calibration

The archaeobotanical assemblage derives from different stratigraphic layers at Skala Sotiros, for which three sets of radiocarbon determinations have been obtained. The first set, obtained from the excavator (Koukouli-Chrysanthaki, personal communication), comprises conventional ¹⁴C dates carried out at the Demokritos laboratory (Athens) on two broad bean (*Vicia faba*) seeds from Trench IIA. In addition, two series of AMS determinations on broomcorn millet (*Panicum miliaceum*) and barley (*Hordeum vulgare*) grains from Trench IIB were carried out at the Poznań Radiocarbon Laboratory (Poland) and at the FTMC Laboratory in Vilnius. Radiocarbon ages (BP) were calibrated in OxCal v4.4.4 (Bronk Ramsey 2021) using the IntCal20 atmospheric calibration curve [62]. All calibrated age ranges correspond to 2σ (95.4%) probability.

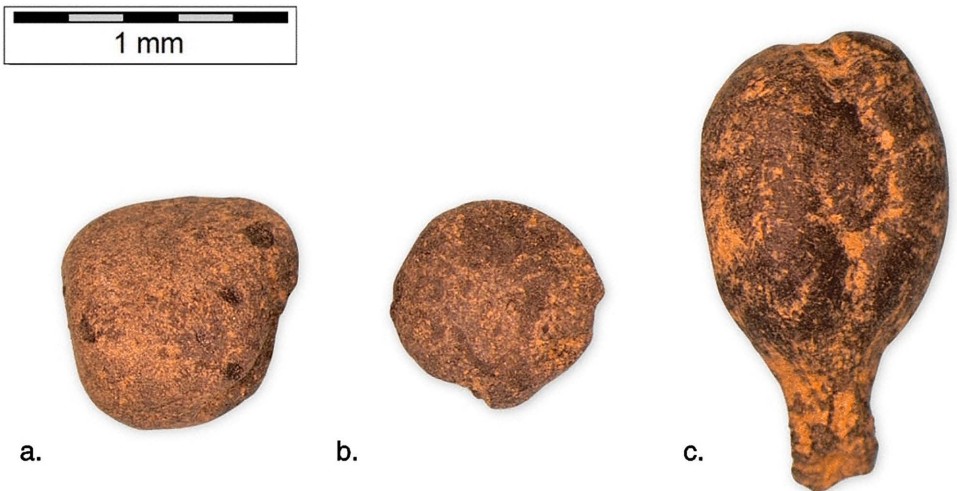

**Fig 3. Charred archaeobotanical remains from Skala Sotiros. (a)** *Lathyrus sativus*, **(b)** *Lens culinaris*, **(c)** *Vitis* sp. Scale bar = 1 mm.

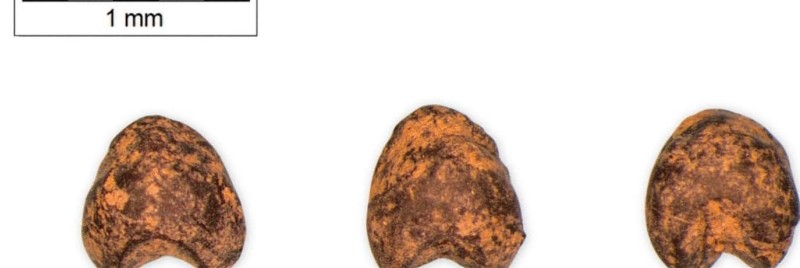

**Fig 4. Charred archaeobotanical remains from Skala Sotiros.** *Panicum miliaceum*. Scale bar = 1 mm.

## Results

### Archaeobotanical results

The archaeobotanical analysis of the Skala Sotiros plant remains has yielded a total of 4664 items from the 69 samples that were sorted and identified (Table 3). The assemblage comprises the remains of cereals, pulses, fruit, and wild/weed taxa that are presented below.

### Cereals

Three cereal species have been identified at Skala Sotiros: millet (*Panicum miliaceum*), barley (*Hordeum vulgare*), and emmer (*Triticum dicoccum*). Overall, cereals are well represented in both absolute numbers and sample frequency. Millet (*Panicum miliaceum*) is the most frequently identified cereal at Skala Sotiros, represented by a total of 533 grains recovered from 38 out of the 69 samples (55.9%). Barley (*Hordeum vulgare*) follows, with 183 grains across 32 samples (46.4%). Of these, 145 grains are well preserved and belong to hulled grains, while 38 grains are more poorly preserved and do not retain sufficient diagnostic features for more detailed morphological classification. Emmer (*Triticum dicoccum*)

**Table 3. Archaeobotanical results from Trenches I, IIA, and IIB at Skala Sotiros.**

| TRENCH | | I | IIA | IIB | TOTAL |
|---|---|---|---|---|---|
| Excavation year | | 1986 | 1987 | 1987 | |
| Number of samples analyzed | | 5 | 33 | 31 | 69 |
| **Cereals** | | | | | |
| *Triticum monococcum/ dicoccum* | grain | – | 5 | 5 | 10 |
| *Triticum monococcum/ dicoccum* | glume base | – | – | 1 | 1 |
| *Triticum dicoccum* | grain | 2 | – | 4 | 6 |
| *Triticum* indet. | grain | 1 | 4 | 1 | 6 |
| *Hordeum vulgare* (hulled) | grain | 1 | 103 | 41 | 145 |
| *Hordeum vulgare* (indet.) | grain | 9 | 21 | 8 | 38 |
| *Triticum/ Hordeum* | grain | 3 | 8 | 4 | 15 |
| Cereal indet. | grain | 1 | 3 | 2 | 6 |
| *Avena* sp. | grain | – | 1 | – | 1 |
| *Panicum miliaceum* | grain | – | 27 | 506 | 533 |
| **Legumes** | | | | | |
| *Vicia faba* | seed | – | 1086 | 1 | 1087 |
| *Vicia ervilia* | seed | – | 3 | – | 3 |
| *Vicia ervilia/Lathyrus sativus* | seed | – | 65 | – | 65 |
| *Lathyrus sativus* | seed | 1 | 854 | 39 | 894 |
| *Lens culinaris* | seed | 2 | 1342 | 115 | 1459 |
| Large legume indet. | seed | 4 | 262 | 59 | 325 |
| **Wild** | | | | | |
| *Rubus* sp. | fruit | – | 1 | – | 1 |
| *Vitis vinifera* L. | seed | 2 | 8 | 3 | 13 |
| *Fumaria* sp. | fruit | – | 1 | – | 1 |
| *Buglossoides arvensis* | fruit | – | 2 | – | 2 |
| *Caryophyllaceae* | seed | – | 1 | – | 1 |
| Poaceae | grain | 5 | 10 | 1 | 16 |
| INDET. | | 3 | 16 | 17 | 36 |
| **TOTAL** | | 34 | 3823 | 807 | **4664** |

is a minor component of the assemblage, identified in 5 samples (7.4%) with a total of 6 grains. The assemblage also contained 10 grains attributed to *Triticum monococcum/dicoccum,* corresponding to cereal grains that could not be distinguished as either einkorn or emmer. A single glume base identified as either einkorn or emmer was found. The near absence of glume bases in the assemblage is likely related to the flotation technique used—specifically the use of cloth instead of fine-mesh sieves, although the possibility of a real absence cannot be ruled out. A small percentage of remains belonged to broader identification categories due to poor preservation of the specimens.

### Pulses

Among the pulses, broad bean (*Vicia faba*) is the most abundant, with 1,087 seeds recovered from 12 samples (17.6%). Lentil (*Lens culinaris*) is the most widely distributed, represented by 1,459 seeds across 28 of the 69 samples (41.2%). Grass pea (*Lathyrus sativus*) occurs in 36 samples (52.9%) with a total of 894 seeds. Bitter vetch (*Vicia ervilia*) is rare, recorded in only three samples (4.4%), each with a single seed. A further six samples (8.8%) contained seeds identified as *Lathyrus sativus/ Vicia ervilia*. In addition, 325 seeds were classified as large legume indeterminate, recorded in 39 samples (57.4%).

### Fruit

Grape (*Vitis vinifera* L.) was the most commonly recorded fruit taxon, represented by 13 seeds from six samples (8.8%). Whether these belong to wild or cultivated varieties cannot be determined due to the lack of diagnostic features. A single seed of *Rubus* sp. was also identified in one sample, possibly representing bramble or raspberry.

### Other wild taxa

A total of 34 wild plant remains were identified in 17 of the 69 samples (25%). These include herbaceous taxa and wild grasses. Specific identifications comprise one fruit of *Fumaria* sp., one seed from the Caryophyllaceae family, and two fruits of *Buglossoides arvensis*, recovered from a single sample. Wild grasses (Poaceae) were recorded in 16 specimens from eight samples (11.8%). Their sparse and scattered presence may reflect background vegetation rather than deliberate use. In addition, 36 carbonized plant remains from 16 samples (23.5%) were classified as indeterminate (INDET.), being too poorly preserved or fragmented for secure taxonomic identification. A single grain of oat (*Avena* sp.) was also recovered from one sample, probably representing a weed. The wild plant assemblage is dominated by taxa with relatively medium to large seeds, while very small-seeded taxa are almost absent, with only a few Poaceae grains present. This pattern further supports the likelihood that small and light remains were underrepresented due to the use of cloth instead of fine-mesh sieves during flotation.

The archaeobotanical assemblage from Skala Sotiros shows marked differences in sample representation across the stratigraphic sequence. The earliest phases (IIa–IIb) are represented by very few samples, while Phase III produced the largest number of samples, resulting in an uneven dataset. For this reason, it is mainly the results of Phase III that are more reliable and are therefore being discussed in greater detail.

Cereal remains are very scarce in the earliest phase (IIa), being represented only by a few barley grains. However, in Phase IIb, millet becomes dominant compared to other cereals, forming dense concentrations in Trench IIb, particularly in the deeper deposits, as well as in final-phase contexts from the upper layers of Phase IIb, associated with the oven and nearby pithoi. Small quantities of barley and rare wheat grains accompany millet, suggesting that these two crops were still cultivated. Their lower proportions in relation to millet in the specific areas excavated may be related to different uses of space within the settlement. They may not reflect the agricultural importance of wheat and barley at the site. Unfortunately, due to the limited area excavated and the relatively small number of samples, it is unsafe to draw generalized conclusions regarding the relative importance of the three cereal species. What is of particular interest, however, is the observation that millet appears during Phase IIb and is absent from the earlier phases at Skala Sotiros. In the later

Phase III, the same pattern continues: millet and barley remain the principal cereals, while wheat is represented only by a few grains recorded sporadically. In Phase III, however, their spatial distribution differs, with contexts in Trench IIB being mainly associated with millet, whereas Trench IIA is characterized by layers richer in barley. Millet and barley occur together in a limited number of samples in Trench IIB, but in most cases, they are found separately.

Pulses, by contrast, are consistently abundant across all phases, though their density and composition vary. Their highest concentrations occur mainly in Trench IIA, especially in burnt layers and storage contexts, whereas in Trench IIB they remain much less frequent. Although the uneven sample representation across phases partly influences this pattern, it nevertheless suggests that lentil, grass pea, and broad bean formed a major and stable component of local agricultural production and diet. Overall, the assemblage from Skala Sotiros reflects a diversified agricultural system centered on millet, barley, and pulses, with millet gaining particular prominence during the later occupation phases. To refine the chronological framework of these phases, a series of AMS radiocarbon determinations was obtained on selected cereal and pulse remains, which are discussed below.

### Radiocarbon results

The radiocarbon dating results (Table 4) provide important additional information on the chronology of the site. Overall, the dates obtained from the two broad bean seeds from Trench IIA fall at the end of the 3rd and the beginning of the 2nd millennium BCE.

Each determination is based on a single grain/seed. Calibrated using OxCal v4.4.4 and the IntCal20 calibration curve.

By contrast, the radiocarbon dates on millet and barley grains from Trench IIB indicate a later phase, falling within the Late Bronze Age, broadly between ca. 1500−1200 cal BCE (95.4%). While stratigraphic and ceramic evidence had originally attributed all analyzed contexts to the Early Bronze Age, the new AMS radiocarbon dates reveal a more complex occupational sequence at Skala Sotiros. The broad bean seeds from Trench IIA date to ca. 2200–2000 BCE, placing them in Phase IIb/c rather than Phase III as previously assumed [55,56]. By contrast, the millet and barley grains from Trench IIB consistently calibrate to the 15th–13th centuries BCE. Two of the three millet determinations from Poznań fall within the range ca. 1400–1200 BCE, while one extends back to around 1500 BCE. For this reason, additional AMS determinations were obtained at the FTMC laboratory in Vilnius on material from the same sample (261). The re-dating of the millet grain produced slightly later results (1409–1219 BCE, 95.4%). Minor discrepancies of c. 100–200 years between the laboratories are not unexpected and can be attributed to differences in laboratory protocols, pretreatment methods, and the intrinsic uncertainties of radiocarbon calibration. Inter-laboratory comparisons, such as the VIRI study [63], have demonstrated that variations of this magnitude fall within the normal range, while continuing revisions of calibration curves (e.g., IntCal20; [62]) further underline the importance of interpreting individual measurements within their broader chronological framework.

**Table 4. Radiocarbon dating results of plant remains from Skala Sotiros.**

| Lab Code | Laboratory | Sample | Material | Radiocarbon Age $^{14}$C (BP) | Calibrated Age (cal BCE, 95.4%) |
|---|---|---|---|---|---|
| DEM–1976 | Demokritos, Athens | Sample 172, Trench IIA | *Vicia faba* | 3723±30 | 2267–2028 |
| DEM–1977 | Demokritos, Athens | Sample 116, Trench IIA | *Vicia faba* | 3720±30 | 2204–2027 |
| Poz-144410 | Poznań, Poland | Sample 260, Trench IIB | *Panicum miliaceum* | 3130±35 | 1497–1294 |
| Poz-144411 | Poznań, Poland | Sample 261, Trench IIB | *Panicum miliaceum* | 3150±35 | 1502–1308 |
| Poz-144412 | Poznań, Poland | Sample 307, Trench IIB | *Panicum miliaceum* | 3115±35 | 1492–1279 |
| FTMC-SW48–5 | FTMC, Vilnius | Sample 261, Trench IIB | *Hordeum vulgare* | 3089±32 | 1427–1266 |
| FTMC-SW48–6 | FTMC, Vilnius | Sample 261, Trench IIB | *Panicum miliaceum* | 3049±34 | 1409–1219 |

The archaeobotanical material examined here formally derives from contexts attributed to the Early Bronze Age, based on the site's material culture and 14C chronology. However, the radiocarbon results indicate two distinct horizons of activity at Skala Sotiros: an Early Bronze Age occupation and indications of Late Bronze Age activity. Specifically, the directly dated millet grains from Skala Sotiros clearly fall in the Late Bronze Age based on the AMS dates from both Poznań and Vilnius laboratories. This apparent inconsistency between the radiocarbon evidence and the absence of material culture securely attributable to the Late Bronze Age cannot be resolved at present, yet it underscores the significance of these new results for refining the chronology of millet cultivation in northern Greece.

## Discussion

The plant species identified at Skala Sotiros are generally consistent with those encountered at other Bronze Age sites in northern Greece. The plant assemblage is characterized by three distinct taxa of pulses and a clear dominance of pulses over cereals. The range of wheat species is considerably narrower, with no evidence for either *Triticum timopheevii* or free-threshing wheats (*T. aestivum/durum*). While einkorn, which occurs in considerable quantities at other sites in northern Greece during both the Neolithic and Bronze Age [63], is not securely attested at Skala Sotiros, barley, well attested regionally [30,63], is also present at Skala Sotiros. This pattern deviates from the general trend observed at neighboring sites in northern Greece, where the assemblages are dominated by cereals and pulses occupy a secondary role in the crop spectrum [30,31].

In contrast, a strong emphasis on pulses has been observed at coastal site Ayios Mamas/Olynthos in northern Greece [64] and at well-studied insular or southern Aegean sites, such as Akrotiri on Thera, where *Lathyrus clymenum* and *L. ochrus* were stored in pure concentrations [65]. This differentiation may reflect island or coastal agro-ecological and dietary orientations during the Bronze Age, while acknowledging the potential influence of sampling quality and recovery strategies at each site.

Among pulses, broad bean (*Vicia faba*, including the so-called Celtic bean), which dominates the assemblage, is absent from Neolithic sites in Greece and appears to have been introduced during the Bronze Age [30]. The Celtic bean reported from Neolithic Arkadikos [33,66] does not correspond to complete and reliably identified Celtic bean seeds. In contrast, lentil, present as a prominent crop at Skala Sotiros, has a long-standing presence since the Neolithic [30].

One of the most notable features of the Skala Sotiros assemblage is the presence of dense concentrations of broomcorn millet. Initially, these finds derived from deep layers attributed to the Early Bronze Age and were therefore considered to date to the early second millennium BCE [32]. However, radiocarbon dating, presented in this paper, clearly shows that the millet from Skala Sotiros belongs to the Late Bronze Age. This discrepancy could be explained by post-depositional intrusion of millet grains into earlier layers, most likely through bioturbation or other taphonomic processes.

Archaeobotanical and radiocarbon evidence from Skala Sotiros and other Bronze Age sites in northern Greece provides crucial insights into the introduction and spread of millet in the Aegean. Previous scholars considered millet to be a native species in Europe due to its presence in Neolithic and Bronze Age contexts [67,68]. However, recent direct dating of millet grains has clearly refuted this view by showing that all 'neolithic' dated millet seeds were intrusive from later habitation layers dated from the Bronze Age onwards [69,70]. It is now generally accepted that millet (both broomcorn and foxtail) is not native to Europe and was introduced to the region around 1500 BCE, gaining popularity in the subsequent centuries [69,70]. Similarly, in Greece, the few millet grains recovered from Neolithic contexts were thought to be intrusive, likely the result of post-depositional disturbance [32].

The earliest directly dated millet grains in the region come from Skala Sotiros, calibrated to 1500–1450 BCE, placing the site among the first localities in South-Eastern Europe [31]. Comparable dates have been obtained from the site of Archontiko [31], where millet grains have been shown to date to 1500–1450 BCE. These findings support the hypothesis that millet reached northern Greece around 1500 BCE [31]. Millet became increasingly common at numerous Late Bronze Age settlements of Greece, suggesting that millet cultivation had become part of the agricultural and food systems of

certain settlements of northern Greece during this period [31,32]. This pattern is attested at sites such as Assiros [28], Kastanas [71], Ayios Mamas/Olynthos [64], Aggellohori [72], and Toumba Thessaloniki [73].

At Assiros [28], the presence of millet exhibits significant variation throughout the Bronze Age. Millet accounted for approximately 16% of the plant assemblage during the site's earlier Late Bronze Age phase (1600–1400 BCE) and rose to 44% in the following phase (1300–1100 BCE). This diachronic pattern reflects a sharp rise in millet's importance during the later stages of the Bronze Age. Isotopic analysis of cattle bones from Phase 5 of the settlement (ca. 1200–1100 BCE), corresponding to the Late Bronze Age-Early Iron Age transition, further supports the importance of millet during this phase, indicating a diet complemented with C4 plants for these animals, with millet as the most likely source [22,74]. At Kastanas, more than 500,000 millet grains were recovered from layers dated between 1600 and 1200 BCE [71]. At Archontiko, only 57 millet grains were recovered from layers dated before 1500 BCE. These most likely represent intrusive material, given that later contexts of the same settlement have yielded over 13,000 millet grains [75,76]. Similar trends are also evident at Ayios Mamas/Olynthos [64], millet appears in low numbers in earlier layers but shows a marked increase in presence during 1440–1340 BCE (phase PO IV of the settlement). According to Kroll [64], this increase is reflected in a sharp rise in presence ubiquity, although earlier layers yielded only individual finds. Archaeobotanical analyses at Aggellochori, a Late Bronze Age site, document a recurrent presence of millet during ca. 1630–1290 BCE. Its representation is generally low to moderate across the assemblages, even when individual samples produce higher counts [72,77]. These findings suggest that millet played an important role in the agricultural practices of the settlement during this period [72]. Further evidence of millet processing comes from both Aggellochori and Archontiko, where grains have been found not only as individual grains  but also as compact fused masses [31].

These data indicate that by the 14th–13th centuries BCE, millet had become a regular component of agricultural production in northern Greece, incorporated into organized storage and consumption systems, as suggested by the finds from Assiros [29] and Toumba Thessaloniki [73]. Millet grains have also been reported from the site of Rema Xydias, located at the foothills of Mount Olympus (Clémence Pagnoux personal communication). However, moving further south to east-central mainland Greece (Phthiotida), detailed archaeobotanical studies at Bronze Age sites yielded no evidence for millet cultivation, despite intensive sampling and systematic analysis of plant macroremains [78].

Although the intensity of archaeobotanical research in southern Greece is relatively lower than in the north, something that is drastically changing over the last years [79–86], the available data from this period nonetheless suggest a clear geographic boundary in the distribution of millet. Secure finds do not extend further south than the Olympus region [31], marking a distinct geographic boundary for the end of the Bronze Age period.

Specifically, in contrast to the northern Greek sites where millet is frequent and, in cases abundant in the Late Bronze Age, assemblages from the southern mainland and Crete provide a different picture for this period. The only clear occurrence of millet from south Greece derives from Tiryns, where ten grains were recovered from Late Bronze Age levels [87]. These could represent imports rather than a palace-grown crop [87], a view consistent with the site's coastal location and its function as an important harbor during the Bronze Age [88]. It is also possible that the grains belong to later phases of occupation beyond the palatial period, since Tiryns continued to be inhabited and even flourished after the collapse of the palatial system, showing substantial rebuilding and architectural investment during the 12th century BCE [88,89]. As such, the association of the millet grains with the palatial phase cannot be securely established without direct radiocarbon dating.

For this period, at other southern sites, millet is either absent or represented only by a few isolated grains from mixed or uncertain contexts. No millet has been identified at Pylos (S. Valamoti, personal communication, based on ongoing analysis), Lerna [90], Midea [91], Tsoungiza [92,93], or Thebes [94]. At Ayios Vasileios [95], Kalapodi [84], and on Crete at Kastelli [96] and Malia [97], there are only a few recorded undated finds (often a single grain) that derive from disturbed deposits (Fig 5). These observations point to the rarity of millet grains from southern Greece during the Bronze Age, which can only be clarified through further research and additional direct radiocarbon dating of the sporadic millet finds from

**Fig 5. Map showing sites with millet finds in the Late Bronze Age layers, Greece.** Map compiled in ArcMap 10.8 using a basemap made from Natural Earth.

the region. In any case, the southern Greek data clearly contrast those from northern Greece, where millet constitutes a significant component of the cereal crop repertoire at most sites analyzed for plant remains.

In parallel with the rise in millet in the north of Greece, an increase in the cultivation of spelt wheat (*Triticum spelta*) is also observed during the Late Bronze Age at some sites of the region. Although spelt was present in Greece since the Early Bronze Age [30,63], archaeobotanical data indicate a clear rise in its frequency and quantity at two sites in northern Greece, particularly during the same phases when millet becomes more prominent. At Kastanas, in northern Greece, spelt cultivation is attested from the Middle–Late Bronze Age and shows an increase in the subsequent LBA levels. [71].

Similarly, at Archontiko, spelt wheat shows a marked increase during the final occupation phase (1500–1400 BCE), where more than 1,000 grains were recovered from 16 samples, in contrast to 375 grains from 4 of 25 samples dating to earlier phases before 1500 BCE [75,76].

Spelt is also attested at other Late Bronze Age sites in northern Greece, including Assiros [28], Ayios Mamas/Olynthos [64], Aggellochori [72], and Toumba Thessalonikis [73], the same sites where millet is documented.

Like millet, spelt wheat is absent from contemporary southern Greek assemblages (Table 5), highlighting significant regional variation in crop selection and cultivation strategies. This pattern reflects a broader shift in agricultural strategies during the Late Bronze Age in northern Greece, marked by greater crop diversity and the introduction or intensification of specific species cultivation [3,31,33,35].

According to Filipović et al. [69] and Martin et al. [101], one possible route for the introduction of millet into Europe originated in the east, from regions of the Caucasus and the Black Sea. The earliest directly radiocarbon-dated grains, found at Vinogradnyi Sad in southwestern Ukraine near the lower Dniester River, date to 1630–1450 BC [69,102]. For northern Greece, directly dated finds from Archontiko (1500–1450 BC; [31]) and Skala Sotiros (1500–1450 BC; 1400–1200 BCE; [31], this study) support the view that one of the routes by which millet entered Europe ran through south-eastern Europe, specifically the Balkans. Moreover, the coastal and insular location of these two sites from northern Greece suggests that maritime communication may have played a crucial role in the process of the spread of millet during the Late Bronze Age [31,32]. Such locations would have facilitated the arrival and dissemination of new crops and practices through coastal exchange networks [31]. The possible parallel operation of both coastal and inland routes in the spread of millet to Europe would have contributed to complex patterns of early millet occurrences and spread rather than a uniform east-west expansion of this crop.

Similar patterns in the introduction of plant species with an eastern origin from Central Asia have been observed for Bronze Age northern Greece. For example, *Lallemantia*, an oil-producing plant that is not native to Greece, was found in concentrations at sites only in northern Greece dating from the Early Bronze Age. The discovery of this species has been associated with trade networks connecting the northern Aegean with Anatolia or the Black Sea, possibly in relation to the trade of tin [35,103].

Beyond plant remains, some archaeological evidence for long-distance contacts between the Black Sea and the Aegean extends back to the Early Bronze Age. The anthropomorphic stelae discovered at Skala Sotiros—reused in later construction phases—have been attributed to the Early Bronze Age and interpreted as reflecting influences from

**Table 5.  Sites in Greece with *Panicum miliaceum* and *Triticum spelta* Late Bronze Age layers.**

| Sites<br>Northern Greece | *Panicum miliaceum* | *Triticum spelta* | References |
|---|---|---|---|
| Aggellochori | X | X | [72] |
| Archontiko | X | X | [75,76] |
| Assiros | X | X | [28] |
| Ayios Mamas | X | X | [64] |
| Kastanas | X | X | [71] |
| Toumba Thessalonikis | X | X | [73] |
| Skala Sotiros | X | – | This study |
| Sites<br>Southern Greece | | | |
| Tiryns | X | – | [87,98] |
| Thebes | – | – | [94] |
| Tsoungiza | – | – | [93] |
| Lerna | – | – | [90] |
| Midea | – | – | [91] |
| Iolkos | – | – | [99] |
| Kalapodi | – | – | [84,100] |
| Pylos | – | – | S. Valamoti, pers. comm. |
| Ayios Vasileios | – | – | [95] |

populations of the Pontic–Black Sea region, where comparable monuments are known [31,55,56,104]. Similar examples from funerary contexts in south-east Bulgaria, site Malomirovo from the Early Bronze Age [105], reinforce this interpretation, pointing to sustained cultural connections across the northern Pontic and European zones.

Human genomic data also support an argument for a Steppe-related ancestry population reaching northern Greece during the early second millennium BCE [106]. This genetic signal demonstrates that interaction networks linking the Pontic steppe with the Balkans and northern Aegean were already in place long before the appearance of millet, providing a possible conduit through which crops from the Steppes could later diffuse. These interactions point to contact and gene flow across the Balkans and northern Aegean, not necessarily involving large-scale population movements [106]. As Anthony [107,108] emphasizes, such evidence does not necessarily point to large-scale migrations but rather to sustained contact through low-level gene flow, marriage alliances, and exchange networks, mechanisms equally capable of transmitting both people and agricultural innovations, including the introduction of new crops such as millet, across regions.

Taken together, these lines of evidence — archaeobotanical, cultural, geographical, and genetic — showcase the northeastern Aegean as a dynamic contact zone bridging the Pontic and Aegean worlds. Through both maritime and overland interaction networks, this region may have served as a conduit for the introduction of millet into Greece during the Late Bronze Age.

In contrast, Steppe-related ancestry has not been detected in individuals from southern Greece [106], underlining the genetic distinction between the northern and southern Aegean during the Bronze Age. Although genomic data from southern Greece do not show the same patterns, nor is there any clear evidence so far for millet cultivation in the region, isotopic analyses from southern Greek skeletal remains [109] indicate that a few individuals consumed significant amounts of C4 plants, probably millet. These cases may represent people of northern origin or displaced individuals who had previously adopted millet-based diets [32].

As regards northern Greece, the introduction of millet coincides with other cultural markers associated with steppe populations [32], including the appearance of horse bones at sites such as Assiros, Kastanas, Aggellohori, and Toumba Thessaloniki. Although bones of domesticated horses have been reported from Early Bronze Age deposits at sites such as Kastanas [110] and Servia [111], their attribution to this period remains uncertain. In Kastanas, the relevant layers may have been disturbed by a Late Bronze Age ditch [112], and the insufficient contextual documentation of the Servia material likewise weakens its chronological reliability [113]. Even if horses were known earlier, their systematic use most likely began during the Late Bronze Age, approximately the same period in which the first securely dated millet grains appear in the archaeobotanical record of northern Greece [32].

Valamoti [32] also notes the spread of a specific type of portable clay hearth, the *pyraunos*, during the same period when both millet and the horse became visible in northern Greece. While such hearths were known earlier in South-Eastern Europe, their frequency increases sharply in Late Bronze Age layers, where they co-occur with millet remains at sites such as Assiros [74], Archontiko [114], and Kastanas [71]. Comparable ceramic forms are notably absent from southern Greek contexts, marking a clear cultural boundary around the Olympus region [12].

The absence of both millet and spelt wheat from southern Greek sites is particularly noteworthy and may reflect a combination of specific plant food choices related to culinary traditions pertaining to different cultural groups as well as distinct agricultural choices or strategies in comparison to their northern counterparts. Although these southern communities maintained contacts with regions of northern Greece, where such crops are well attested, they appear to have remained committed to their already existing crop choices and did not readily adopt them. The absence of millet is especially notable given the broader environmental pressures of the period. Climatic stress associated with the so-called 3.2 ka event, the climatic downturn broadly centered in the 12th century BCE, brought episodes of severe drought and agricultural instability to the Mediterranean. Even if these climatic conditions did not directly lead to the abandonment of the Mycenaean palatial centers, they may have contributed to episodes of significant drought and agrarian disruption both across the wider Mediterranean and the Greek mainland. Despite the fact that millet is a drought-tolerant cereal with low input requirements

and potential use as a rotation crop, it was not adopted in southern Greece and the Mycenaean world. This may indicate cultural resistance to new crops and agricultural practices. According to Valamoti [31,32], plant choices in Greece and cooking practices are closely linked to cultural identity; in this context, the exclusion of both millet and the *pyraunos* from southern Greek culinary practices may represent deliberate choices of differentiation from the northern communities that used them. Beyond questions of identity differentiation, it is also important to consider how structural constraints within the prevailing economic and political systems may have limited the adoption of new crop species [115–117].

In the Greek case, the uneven adoption of millet (and other crops) may suggest that localized, bottom-up dynamics in the north facilitated experimentation, while top-down control in the south may have constrained it. In southern Greece, Mycenaean palatial centers (e.g., Mycenae, Pylos, Thebes, Tiryns) oversaw centralized agricultural production under bureaucratic control, focusing on staple crops such as wheat, barley, wine, and olives [7,15,118]. These elite-controlled systems relied on established agricultural practices and the generation of agricultural surplus to sustain redistributive economies [119]. As such, there may have been little incentive, or even institutional resistance, to altering existing practices, especially when innovation could risk disrupting a well-functioning system [117,120].

In contrast, northern Greece was characterized by smaller, autonomous communities practicing decentralized, household-based farming [4,25,26], which may have allowed for greater flexibility and openness to agricultural innovation, including the adoption of new crops such as millet. Even in the case of Assiros, where surplus storage facilities have been documented, the scale and organization of agricultural production remain fundamentally different from the palatial economies of southern Greece. This reinforces the distinction between top-down systems—characterized by elite-driven, large-scale production—and more flexible, bottom-up models that allowed for local decision-making and the incorporation of new crops like millet.

Northern Greek communities of the Bronze Age, more exposed to trans-Balkan and Steppe networks, appear to have followed distinct developmental trajectories shaped by their geographic position and sustained contact with regions beyond the Aegean. The flexibility, visible in the integration of new crops and practices, may have contributed to the region's smoother transition from the Late Bronze Age to the Early Iron Age, as evidenced by continuity in archaeological records [26].

Recent research in Central Europe (e.g., Tiszafüred-Majoroshalom, Hungary) has demonstrated population movement during the Late Bronze Age directly associated with the introduction and increasing importance of millet [121]. For Greece, a similar scenario can be hypothesized: millet may have been introduced to northern Greece not merely as a traded commodity but as part of the subsistence practices of populations moving southwards from the east. This interpretation accords with genomic evidence showing Steppe-related ancestry in northern but not southern Greek populations [106]. In addition, or parallel to existing contact networks, it is equally likely that new nomadic groups who already used millet as an integral part of their agropastoral economies and nomadic lifestyles, to which millet is ideally suited due to its short life-cycle, moved to southeastern Europe and the north Aegean.

If millet arrived with such populations, its subsequent cultivation in northern Greece reflects both environmental suitability and cultural transmission through mobility and interaction. The boundary south of Mount Olympus indicates that millet's diffusion was mediated more by social and ideological mechanisms than by ecological constraints as regards the south Greece. Ultimately, the introduction of a crop does not guarantee its adoption: *cultural selection*, the acceptance or rejection of innovations according to local norms, identities, and institutions, played a decisive role. In Greece, millet became embedded in flexible, community-based systems of the north, while palatial economies of the south appear not to have adopted it. The resulting uneven distribution of millet across Greece encapsulates broader patterns of social differentiation and cultural filtering that shaped agricultural practice and identity during the Late Bronze Age.

## Conclusions

The archaeobotanical evidence from Skala Sotiros provides direct insights into agricultural development in northern Greece during the Bronze Age, revealing a shift in plant economy marked by the increased presence of broomcorn

millet (*Panicum miliaceum*); radiocarbon dating from Skala Sotiros, together with other directly dated millet remains from Greece, confirms that the cultivation of this crop was already part of the agricultural system of northern Greek communities by the Late Bronze Age.

Taken together with the broader archaeobotanical and radiocarbon evidence from other Greek sites reviewed in this study, the results indicate that northern Greek communities were more receptive to agricultural innovation and more flexible in their subsistence strategies compared with their southern counterparts. Communities in the north adopted new crops, such as millet, which were likely introduced through contacts with the Balkans and the Pontic Steppe-Black Sea. The rise in aridity at 1200 BCE prompted more ubiquitous use of millet in northern Greece, thus reflecting agricultural resilience. However, the direct impact of climate on agricultural systems requires further investigation.

This research highlights regional variation in agricultural practices in Greece and emphasizes the nuanced understanding of Bronze Age subsistence, mobility, cultural exchange, culinary practices, and resilience to climate stress, while illustrating key differences between northern and southern Greece.

## Supporting information

**S1 File. Archaeobotanical dataset.** Excel file containing the full archaeobotanical dataset used in this study.
(XLSX)

**S2 File. Inclusivity in global research checklist.** Completed PLOS checklist.
(PDF)

## Acknowledgments

We would like to thank the excavator of the site, Chaido Koukouli-Chrysanthaki, for providing valuable information on the excavation and for granting access to the original excavation plans and the archaeobotanical samples from Skala Sotiros. We also thank Marianna Tomara for assisting the authors with the maps.

## Author contributions

**Conceptualization:** Kyriaki Karanikola.

**Data curation:** Kyriaki Karanikola, Soultana-Maria Valamoti.

**Formal analysis:** Kyriaki Karanikola.

**Funding acquisition:** Soultana-Maria Valamoti, Giedre Motuzaite Matuzeviciute.

**Investigation:** Kyriaki Karanikola.

**Methodology:** Kyriaki Karanikola, Soultana-Maria Valamoti, Giedre Motuzaite Matuzeviciute.

**Supervision:** Soultana-Maria Valamoti, Giedre Motuzaite Matuzeviciute.

**Validation:** Kyriaki Karanikola.

**Visualization:** Kyriaki Karanikola.

**Writing – original draft:** Kyriaki Karanikola.

**Writing – review & editing:** Kyriaki Karanikola, Soultana-Maria Valamoti, Giedre Motuzaite Matuzeviciute.

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
