## [Decision Letter · Decision Letter 0]

22 Dec 2025

Dear Dr. Karanikola,

All three reviewers are positive about your paper but raise substantial comments regarding its content and methodological approaches. In light of these reviews, the manuscript requires some revision. Please take all of these comments into careful consideration when revising your manuscript.

We look forward to receiving your revised manuscript.

Kind regards,

Stefanos Gimatzidis, Ph.D.

Academic Editor

PLOS One

Journal Requirements:

3. In your manuscript, please provide additional information regarding the specimens used in your study. Ensure that you have reported human remain specimen numbers and complete repository information, including museum name and geographic location.

For more information on PLOS One's requirements for paleontology and archeology research, see https://journals.plos.org/plosone/s/submission-guidelines#loc-paleontology-and-archaeology-research .

4. Please note that funding information should not appear in any section or other areas of your manuscript. We will only publish funding information present in the Funding Statement section of the online submission form. Please remove any funding-related text from the manuscript.

“This research was funded by the European Research Council (ERC) under the European Union’s Horizon 2020 research and innovation programme (Consolidator Grant No. 101087964, project MILWAYS, PI Giedrė Motuzaite-Matuzeviciute).”

6. In the online submission form, you indicated that “All relevant data supporting the findings of this study are available within the manuscript and its Supporting Information files. Additional raw data, including radiocarbon determinations and archaeobotanical counts, are archived at Vilnius University and at Aristotle University of Thessaloniki and can be made available upon reasonable request.”

7. We note that Figures 1,2, 3 and 6 in your submission contain map/satellite images which may be copyrighted. All PLOS content is published under the Creative Commons Attribution License (CC BY 4.0), which means that the manuscript, images, and Supporting Information files will be freely available online, and any third party is permitted to access, download, copy, distribute, and use these materials in any way, even commercially, with proper attribution. For these reasons, we cannot publish previously copyrighted maps or satellite images created using proprietary data, such as Google software (Google Maps, Street View, and Earth). For more information, see our copyright guidelines: http://journals.plos.org/plosone/s/licenses-and-copyright.

1. You may seek permission from the original copyright holder of Figures 1,2, 3 and 6  to publish the content specifically under the CC BY 4.0 license.

8. Please include your tables as part of your main manuscript and remove the individual files. Please note that supplementary tables (should remain/ be uploaded) as separate "supporting information" files.

Reviewers' comments:

Reviewer's Responses to Questions

**Comments to the Author**

1. Is the manuscript technically sound, and do the data support the conclusions?

Reviewer #1: Yes

Reviewer #2: Yes

Reviewer #3: Yes

2. Has the statistical analysis been performed appropriately and rigorously?

Reviewer #1: N/A

Reviewer #2: Yes

Reviewer #3: Yes

3. Have the authors made all data underlying the findings in their manuscript fully available?

Reviewer #1: Yes

Reviewer #2: Yes

Reviewer #3: Yes

4. Is the manuscript presented in an intelligible fashion and written in standard English?

Reviewer #1: Yes

Reviewer #2: Yes

Reviewer #3: Yes

Reviewer #1: Dear authors, you have produced a very well thought out and well-written contribution to the early story of Panicum miliaceum in Europe. I am looking forward to see it published - and actually I have nothing to be added or chaqnged in its contents. This is rare and makes me very happy.

Still, I would kindly request modifications in terms of readability:

1. The section on radiocarbon dates is extremely difficult to read in the current state. As you have perfectly layouted tables, I do not see the necessity to repeat so many of them in the text.

2. You submitted all the tables as supporting information, and I strongly advise against this choice. The tables are small, highly informative, and in my opinion they would all perfectly fit into the document itself.

3. Please consider merging tables 4, 5, and 6 into one table.

Reviewer #2: I very much enjoyed reading this paper. It is good, solid research that is very well contextualised and provides significant new insights not only on the dispersal of millet but also on cultural choices of food. I suggest its publication with some minor revisions as follows:

Intro: why not include mentions on Crete as it is really part of this Mycenaean-Minoan framework?

Data ARE (and not ‘is’)

Results -Cereals:

‘Four cereal species have been identified at Skala Sotiros: millet (Panicum miliaceum), barley 255 (Hordeum vulgare), emmer (Triticum dicoccum), and einkorn (Triticum monococcum).’: from the discussion below and the table it seems that it is not clear if einkorn was present (T. mono/dico) so maybe remove it from the opening sentence or edit to reflect this. You also mention einkorn (as secure presence) in the discussion, so this needs editing too.

Not sure how you differentiate between H. vulgare and Hordeum sp. All barley should be H. vulgare and then you have subspecies for hexastichum or distichum. Do you mean H. vulgare as in 6-row barley? Please clarify as the different ways of nomenclature may cause confusion.

In regards to the mesh size and its potential impact: what about the wild taxa that you found? Did you find only bigger ones or did you also have small seeds? This can give you another indication on whether you lost material or not. As far as I can see most are bigger ones but maybe some of the Poaceae ones are small? Using this line of evidence you can make a comment (in all likelihood that material was lost).

Results overall: I find it too wordy; there is a lot of detail on the finds of each trench that is not needed. I think this information can be best summarised in a table and discuss the finds in a more concise manner in the text, which you do anyway below when you synthesise the finds, after the wild taxa. I recommend deleteing many of the details on descriptions and extend instead the discussion of the general trends of the results across period and area.

Nomenclature: you may want to use updated nomenclature. Please choose an (ideally updated) flora and add in the text which nomenclature you are following. Graminae for example are now named Poaceae, etc etc.

Line 362: ‘burning layers’ change to ‘burnt layers’

Radiocarbon results: some of this information needs to be moved to the methods section and leave here only the results.

In regards to the RC results and the discrepancy with archaeological data: it is worth considering and discussing in the text whether any of the grains (especially in the case of millet) could be later intrusions into older layers due to bioturbation or other potential taphonomic factors.

Discussion:

presence frequency (Stetigkeit): I believe you refer here to ‘ubiquity’?

Reviewer #3: Earliest millet cultivation reflects steppe connections, dietary flexibility,

and resilience in Bronze Age northern Greece

A valuable paper, very informative and well written. As far as I can judge the data is sufficiently documented, the discussion is conceptually well-developed, and the conclusions are scientifically valid. I have some (few) proposals that might further enhance the acceptance of the paper by collegues.

Proposed Corrections

Around Line 70-71:

In context with your discussion of Middle and Late Bronze Age Networks in the Aegean [and especially since you mention that “Mycenaean influence in northern Greece is only minimally attested and mainly confined to a few ceramic finds”], you may wish to discuss/reference the recent (2024) large-scale research at Adatepe, by Reinhard Jung and Hristo Popov (eds):

“Searching for Gold-Resources and Networks in the Bronze Age of the Eastern Balkans”. The complete volume is available for download: austriaca.at/0xc1aa5572 0x003f5d96.pdf

Around Line 91 (ff)

In context with your discussion of climate and environmental change during the Bronze Age,

and especially since you mention that “two major climatic events” (at 4.2 ka calBP and 3.2 ka calBP) are under discussion, you may like to add the proposal by Bernhard Weninger, Eelco Rohling and collegues that societally significant climate variability may also have occurred in the Aegean, and particularly in the Northern Aegean, in the transition from the Late Bronze Age to the Early Iron Age (sometimes called “Dark Ages”) at around 1050-980 BCE:

Weninger, B., Clare, L., Rohling, E.J. Bar-Yosef, O.,l Böhner, U., Budja, M., Bundschuh, M., Feurdean, A., O., Linstädter, O., Mayewski, P., Muhlenbruch, T., Reingruber, A., Rollefson, G., Schyle, D., Thissen, L., Zielhofer, C., 2009. The Impact of Rapid Climate Change on prehistoric societies during the Holocene in the Mediterranean. Documenta Praehistorica XXXVI, 7-59. DOI: 10.4312/dp.36.2

Rohling, E.J., Marino, G., Grant, K.M., Mayewski, P.A., Weninger, B., 2019.A model for archaeologically relevant Holocene climate impacts in the Aegean-Levantine region (easternmost Mediterranean). Quaternary Science Reviews, Volume 208, 2019, Pages 38-53, ISSN 0277-3791. https://doi.org/10.1016/j.quascirev.2019.02.009.

You might also like to add the (in my view: useful) overview of the available climate records in the Aegean (and Near Eastern) Bronze Age by:

Jacobson, M. J., Seguin, J., & Finné, M. (2024). Holocene hydroclimate synthesis of the Aegean: Diverging patterns, dry periods and implications for climate-society interactions. The Holocene, 0(0). 1-17 https://doi.org/10.1177/09596836241275028

Lines 390-391

You might like to abbreviate the repetitive notation of the calibrated ages by the following

method or similar:

Sample 260 (Poz-144410) produced an age of 3130 ± 35 BP: 1497–1294 cal BCE (95.4%).

**Do you want your identity to be public for this peer review?** For information about this choice, including consent withdrawal, please see our Privacy Policy

Reviewer #1: No

Reviewer #2: **Yes:** Alexandra Livarda

Reviewer #3: No

---

## [Author Response · Author response to Decision Letter 1]

27 Jan 2026

Please see the detailed, point-by-point Response to Reviewers provided as a separate file. All comments from the Academic Editor and the reviewers have been addressed in the revised manuscript.

---

## [Editor Report · Decision Letter 1]

9 Feb 2026

Dear Dr. Karanikola,

Thank you for the revised version of your paper. Upon recent review, I noticed that Table 1 (Archaeological Phases and Chronology for Northern Greece) still relies on outdated chronological data regarding the transition from the Late Bronze Age (LBA) to the Early Iron Age (EIA).

In light of recent analytical studies at sites such as Assiros and Sindos, this transition is now placed within the 12th century BCE. This discrepancy persists throughout your manuscript, particularly concerning the dating of the Assiros stratigraphic sequence and general Aegean periodization.

Please revise the text and the table to align with current high-precision chronologies, specifically incorporating the findings from the following works:

https://journals.plos.org/plosone/article?id=10.1371/journal.pone.0232906

https://journals.plos.org/plosone/article?id=10.1371/journal.pone.0106672
Gimatzidis, S. 2024. Chronological revision in the Aegean: Perceptions of time along stratigraphic and ceramic sequences, in: J. Driessen and T. Fantuzzi (eds), Chronos. Stratigraphic Analysis, Pottery Seriation and Radiocarbon Dating in Mediterranean Chronology. Aegis 26, 321–50. Louvain: Presses universitaires de Louvain.

We look forward to receiving your revised manuscript.

Kind regards,

Stefanos Gimatzidis, Ph.D.

Academic Editor

PLOS One
---

## [Author Response · Author response to Decision Letter 2]

12 Feb 2026

For the response to the comment, find the attached file: Response to Reviewers.

---

## [Editor Report · Decision Letter 2]

16 Feb 2026

Earliest millet cultivation reflects steppe connections, dietary flexibility, and resilience in Bronze Age northern Greece

PONE-D-25-57342R2

Dear Dr. Karanikola,

We’re pleased to inform you that your manuscript has been judged scientifically suitable for publication and will be formally accepted for publication once it meets all outstanding technical requirements.

Kind regards,

Stefanos Gimatzidis, Ph.D.

Academic Editor

PLOS One
---

## [Editor Report · Acceptance letter]

PONE-D-25-57342R2

PLOS One

Dear Dr. Karanikola,

I'm pleased to inform you that your manuscript has been deemed suitable for publication in PLOS One. Congratulations! Your manuscript is now being handed over to our production team.

Kind regards,

on behalf of

Dr. Stefanos Gimatzidis

Academic Editor

PLOS One